# Evaluating physiological responses and forward head-neck posture in primary students in Thailand when carrying modified and traditional backpacks

Jittaporn Mongkonkansai[1], Udomsak Narkkul[2], Chadapa Rungruangbaiyok[3], Chuchard Punsawad[2]*

1 College of Graduate Studies, Excellence Center for Public Health Research, School of Public Health, Walailak University, Nakhon Si Thammarat, Thailand, 2 Department of Medical Science, School of Medicine, Center of Excellence in Tropical Pathobiology, Walailak University, Nakhon Si Thammarat, Thailand, 3 Department of Physical Therapy, School of Allied Health Sciences, Movement Science and Exercise Research Center, Walailak University, Nakhon Si Thammarat, Thailand

* chuchard.pu@wu.ac.th

## Abstract

Carrying backpacks daily imposes a physical burden on school-aged children, raising increasing public concern. This study compared the physiological responses and head–neck posture of primary school students when carrying traditional versus modified bags. Thirty-two public primary school students aged 7–12 years participated. Physiological measures included heart rate (HR), oxygen saturation ($SpO_2$), blood pressure, and respiratory rate (RR). Head and neck tilt angles and forward head posture were assessed using photogrammetry and Kinovea software. Participants walked at a natural pace (1.1–1.3 m/s) along an 8-meter walkway for five minutes under five conditions: no bag, bilateral traditional bag, unilateral traditional bag, bilateral modified bag, and unilateral modified bag. Backpack design showed significant main effects on heart rate ($p = 0.001$, $\eta p^2 = 0.316$), respiratory rate ($p = 0.007$, $\eta p^2 = 0.214$), systolic blood pressure ($p < 0.001$, $\eta p^2 = 0.387$), and diastolic blood pressure ($p = 0.002$, $\eta p^2 = 0.279$), with higher values in the traditional condition (e.g., HR 95.94 ± 2.26 bpm; RR 31.59 ± 0.86 breaths/min; systolic 108.33 ± 1.30 mmHg; diastolic 78.05 ± 1.64 mmHg). Carrying method affected HR ($p = 0.014$, $\eta p^2 = 0.178$), with higher values during unilateral use. No interaction effects were found for physiological variables ($p > 0.05$). $SpO_2$ remained stable (98.78–99.47%). For posture, significant effects of design were observed for neck tilt ($p < 0.001$, $\eta p^2 = 0.350$), head tilt ($p = 0.001$, $\eta p^2 = 0.324$), and forward head posture ($p < 0.001$, $\eta p^2 = 0.335$). The traditional backpack produced smaller neck tilt (49.75° ± 1.05°) but greater head tilt (78.60° ± 1.56°) and forward head posture (8.68 ± 4.59 cm). Unilateral carrying increased head tilt (77.83° ± 1.56°, $p = 0.021$). Interaction effects were found for neck and head tilt ($p < 0.05$). The modified backpack reduces physiological strain and

**Data availability statement:** All data associated with this study have been included in this article.

**Funding:** This project was funded by the National Science and Technology Development Agency (NSTDA) of Thailand and the Walailak University Graduate Research Fund (CGS-RF-2023/04).The funders had no role in study design, data collection and analysis, decision to publish, or preparation of the manuscript.

**Competing interests:** The authors have declared that no competing interests exist.

improves postural alignment, particularly under unilateral carrying, while bilateral use minimizes asymmetrical loading.

## Introduction

Carrying a school bag is a routine activity for most children [1], and 90% of schoolchildren in developed countries use a backpack [2]. Many of them carry backpacks weighing between 10% and 20% of their body weight [3], with the contents typically including laptops, textbooks, pencil cases, scientific calculators, sports uniforms, school uniforms, lunch boxes, and filled water bottles [4]. Such load carriage can negatively affect children's musculoskeletal systems, potentially leading to headaches, muscle strain, and chronic pain in the back, neck, and shoulder [5].

Numerous studies have demonstrated that carrying heavy school bags can lead to a reduction in walking speed [6,7] and greater force exertion on the spinal intervertebral discs [8], which can affect postural alignment. Research has shown that carrying a backpack affects the cervical posture, leading to a reduced craniovertebral angle (CVA) and impaired pulmonary function [9]. As backpack loads increase, children exhibit a progressive decrease in CVA, trunk forward lean angle, and pelvic tilt angles [10]. Notably, reduced CVA suggests craniovertebral protrusions. Additionally, pain and discomfort intensify with backpack carriage [10], with significantly larger cranial angles to the horizontal, while the craniovertebral angle and sagittal shoulder posture are notably reduced when carrying a traditional backpack compared with carrying no load [11].

Previous studies have examined the proportion of backpack dimensions relative to the student's body size [12,13], comparing the effects of various school bag designs, including a traditional school bag, and the impact of bilateral strap versus unilateral single strap carriage on body balance and neck angle in children. These findings indicated that carrying the bag using bilateral straps helped children maintain body balance and a normal neck position, similar to those observed in children not carrying any load [11]. This has led to the development of backpack harnesses designed to achieve better outcomes and reduce back pain among students [14].

Backpack use is prevalent among both adults and children and is frequently associated with the development of musculoskeletal discomfort. Previous studies have demonstrated that when the load is carried on a single shoulder, the ability to control pelvic rotation and obliquity decreases [15]. Moreover, using a one-strap backpack leads to increased activation of the trapezius muscle and greater discomfort in the neck and shoulder region [16]. To address this issue, new designs have been developed aimed at optimizing straps, including wider straps and additional support straps across the chest, waist, and hips [17–19]. The discrepancy between the size and weight of a child's backpack represents a significant concern for the health of school-aged children. Researchers have suggested that school backpacks should include padding in the back and shoulder areas and that straps should be adjustable to accommodate children's anthropometric measurements [20]. Heart rate (HR), blood pressure (BP), oxygen saturation ($SpO_2$), and respiratory rate (RR) are the primary

vital signs. Assessing these measurements serves as the foundation for evaluating both health and overall well-being and can also indicate physical responses [21,22], with some researchers suggesting that carrying a heavy bag leads to elevated cardiorespiratory responses and an overall increase in physiological stress [23]. This study aimed to explore the effect of a new school bag developed for children, which considers physiological and biomechanical factors, when compared to a traditional backpack, no backpack, and bilateral versus unilateral backpack carriage. This study presents a newly designed bag and aims to evaluate physiological responses and forward head-neck posture in comparison to a traditional bag, with the goal of mitigating the harmful effects of the carried load and minimizing the potential adverse impacts of its use.

## Materials and methods

### Study design and site

An experimental study was conducted on primary public-school students, typically aged 7–12 years, who were enrolled in grades 1–6 during the 2024 academic year at public schools in Muang, Nakhon Si Thammarat, Thailand.

### Participants

Participant recruitment was carried out from 28 October to 29 November 2024. The study protocol was reviewed and approved by the Human Research Ethics Committee of Walailak University (approval no.: WUEC-24-004-01). Written informed consent was obtained from each participant's parent or legal guardian for participation and for the use of any potentially identifiable images or data. The sample size for the study was determined following the Heinisch study [24]. The sample size was calculated using the formula ($n = [(Z\alpha + Z\beta)^2 \times \sigma^2]/ (\mu_0 - \mu)^2$). In this formula, n is the number of participants required, $Z\alpha$ = level of confidence ($\alpha$ = 0.05, $Z\alpha$ = 1.96), $Z\beta$ = power of test (set at power = 95%, $\beta$ = 0.05, so $Z\beta$ = 1.645), $\sigma^2$ = standard deviation difference [19], $\mu_0 - \mu$ = variation in average muscular activity across compartments weighing 15% of body weight [19]. Consequently, the minimum sample size was 10. To account for potential sample loss, the sample size was adjusted using a method that ensures the acquired data have sufficient magnitude to avoid a significant impact on the standard error. The sample size was calculated using the formula n_adj = n/(1 − d)². In this formula, n_adj represents the adjusted sample size, n is the initial sample size, and d denotes the proportion of missing or lost data expressed as a percentage (the loss rate was set at 35%). Based on this calculation, the required sample size for the study was 34 students. Participants were selected through a multistage random sampling process, stratified by grade level and gender. However, due to technical issues with the equipment that led to incomplete data collection, only 32 complete datasets were obtained from the intended 34 participants.

Children recruited for this study were asked if they wished to participate and received all relevant research information before data collection began. Written informed consent was obtained from all parents and guardians, and assent was obtained from all children before data collection. All collected data were anonymized and stored in a locked filing cabinet.

The inclusion criteria for both sexes included individuals aged 7–12 years with no current back or lower limb pain that could interfere with walking movements. The exclusion criteria were as follows: intellectual disabilities, epilepsy, cerebral palsy, hemiparesis, diabetes mellitus, heart and circulatory diseases, neurovascular and congenital problems, respiratory system disorders, a history of fractures or injuries to the lower extremities within the past year, serious back pain in the previous 3 months, newly confirmed radiological fractures or limb injuries, and a history of surgery during the last 6 months.

### Data collection

**Anthropometric data.** The students' anthropometric measurements were obtained using a standard anthropometric instrument (Anthropometer-Serial No.052855, TTM, Japan). The measurements included stature height, shoulder-elbow height, sitting shoulder height, sitting elbow height, hip breadth, shoulder breadth, waist circumference, chest circumference, and body weight, followed by a literature review [25,26]. The instrument included a weight scale, a

stadiometer, a measuring tape, a sliding caliper, and a sliding C-shaped arm. The measurements were taken three times, and the average was calculated and recorded in centimeters.

**Physiological response.** The following physiological responses were assessed: peripheral oxygen saturation ($SpO_2$), heart rate (HR), systolic blood pressure (SBP), diastolic blood pressure (DBP), and respiratory rate (RR).

Peripheral oxygen saturation and Heart rate. HR and $SpO_2$ were measured using a pulse oximeter (Rad-5V, Masimo SET, Serial No. 0000174022, CA 92618, USA) by placing the index finger on the probe. The device was compliant with ISO 80601-2-61 standards, ensuring validated accuracy [27]. Participants were seated comfortably, with the probe placed on the fingertip. Care was taken to minimize movement and ambient interference. Readings were recorded only when values remained stable within the acceptable range for at least 10 seconds [28].

Blood pressure. Blood pressure was measured using a digital blood pressure monitor (IP21, RoHS, Serial No. 230915, China), following the American Heart Association (AHA) protocol [29–31]. Participants refrained from vigorous physical activity for at least 30 minutes before measurement. They were seated in a relaxed position, with arms supported on the back of a chair or table. Measurements were taken at 5-minute intervals. The cuff (22–26 cm arm circumference) was fitted snugly around the upper arm, approximately 2–3 cm above the antecubital fossa. Two readings were taken at 5-minute intervals. If the difference between the two exceeded 5 mmHg, a third reading was obtained, and the average of the two closest values was used for analysis. Systolic and diastolic blood pressure were recorded in mmHg, with the mean value taken as the result.

Respiratory rate. Respiratory rate was measured manually by counting visible chest wall movements for 60 seconds while participants were at rest. Each complete inspiration–expiration cycle was recorded as one breath, following World Health Organization guidelines [32]. Manual counting of respiratory rate is regarded as the clinical gold standard, although it is subject to observer variability [33,34].

**Forward head-neck posture.** A photographic method from the lateral view was used to evaluate each subject's posture using adhesive markers affixed to the seventh cervical vertebra, tragus of the ear, and canthus anatomical landmarks [35]. Photographs were taken using a Canon ES600 digital camera, mounted on a tripod positioned 0.8 meters in front of the participant, with the lens axis aligned perpendicular to the sagittal plane and at a height corresponding to the C7 spinous process [36]. Head-neck posture was analyzed using Kinovea Software version 2023.1.2 [37]. The following measurements were obtained: head tilt angle, defined as the angle between the line connecting the tragus (the cartilaginous projection anterior to the auditory meatus) and the canthus (the lateral fissure of the eyelid) and the vertical axis; neck tilt angle, defined as the angle produced between the line connecting the seventh cervical vertebra (C7) and the tragus of the ear and the horizontal; and forward head shift, defined as the horizontal distance between the tragus and the seventh cervical vertebra. The forward head-neck posture measurement was shown in Fig 1.

**School bag.** The bag load was set at 10% of each student's body weight, and students carried the bags using either bilateral or unilateral straps. This study examined two types of school bags: a traditional bag and a modified bag. The traditional bag represented the standard school-issued backpack, provided by the school and uniform in style across all students. It featured two shoulder straps, a back pocket, and small mesh side pockets for water bottles. The dimensions of the traditional bag were 25×45×15 cm (width×length×thickness) (Fig 2A). The modified bag was newly designed based on anthropometric data of students and findings from previous studies [3,19,38]. It featured two shoulder straps, a back pocket, chest and waist straps, and nano-rubber padding integrated into the back panel and straps. The dimensions of the modified bag were 20×40×5 cm (width×length×thickness), with strap widths of 7 cm and adjustable lengths ranging from 53–92 cm (Fig 2B).

## Procedure

The procedure was divided into three steps: baseline data collection, measurements during the different conditions, and measurements immediately post each carriage condition.

Baseline physiological measurements were obtained before walking under each condition. Parameters included HR, $SpO_2$, SBP, DBP, and RR. Participants were also asked to report their levels of discomfort, and head-neck posture

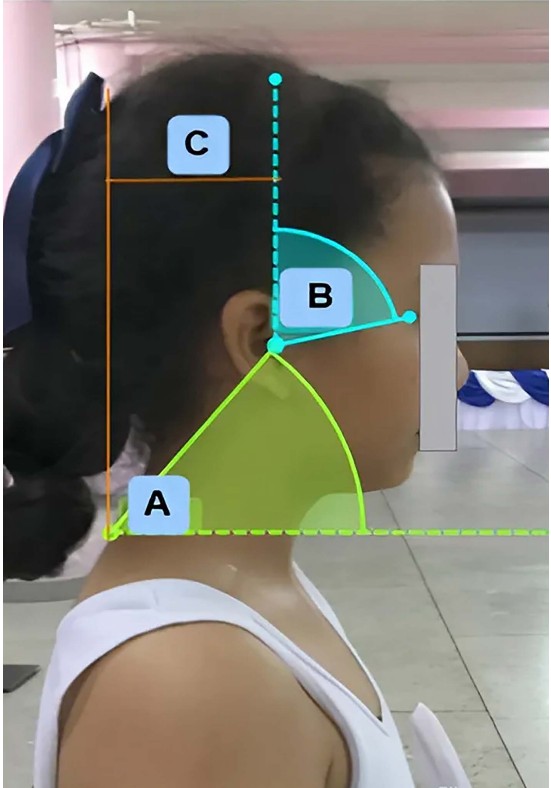

**Fig 1. Forward head-neck posture measurement. (A)** neck tilt angle. (**B**) head tilt angle. (**C**) forward head distance.

was assessed. Individuals were then tested under five conditions: no bag, bilateral traditional bag, unilateral traditional bag, bilateral modified bag, and unilateral modified bag (Fig 3). Each participant was asked to walk along an 8-meter walkway at a comfortable pace (1.1–1.3 m/s) for 5 min, with a 10% body weight load in all backpack conditions. The researchers demonstrated the proper walking pace, instructing participants to walk naturally without rushing or slowing down to ensure a clear understanding of the procedure before the actual test. The sequence of conditions was randomized, and a 5-minute rest period was provided between each condition. Before commencing the subsequent trial, physiological measures and muscle activity were assessed to ensure that all parameters had returned to baseline values. Moreover, both loads were placed in identical bags throughout the experiment, and participants were blinded to the order in which they carried each bag. Upon completion of each task, physiological measures and head-neck posture were recorded.

## Statistical analysis

Data were collected and processed using SPSS version 25 (IBM Corp., Armonk, NY, USA). Data were tested using Shapiro Wilk tests and found to be normally distributed and suitable for parametric statistical analysis. Descriptive data were calculated using frequency, percentage, mean, and standard deviation (SD). A two-way repeated measures ANOVA was conducted to examine the main effects of bag type (traditional vs. modified) and carrying method (unilateral vs. bilateral), as well as their interaction effect, on physiological responses and forward head–neck posture variables. When significant effects were identified, post hoc pairwise comparisons were performed using paired t-tests for predefined contrasts based on the study hypotheses. Statistical significance was set at $p < 0.05$.

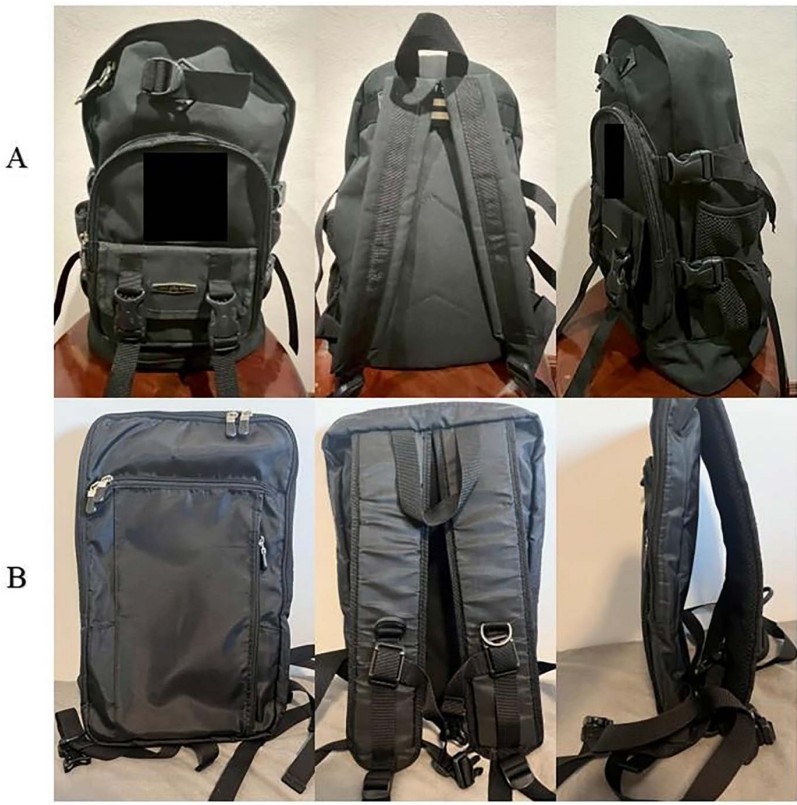

**Fig 2. Type bag; A = traditional bag, B = modified bag.**

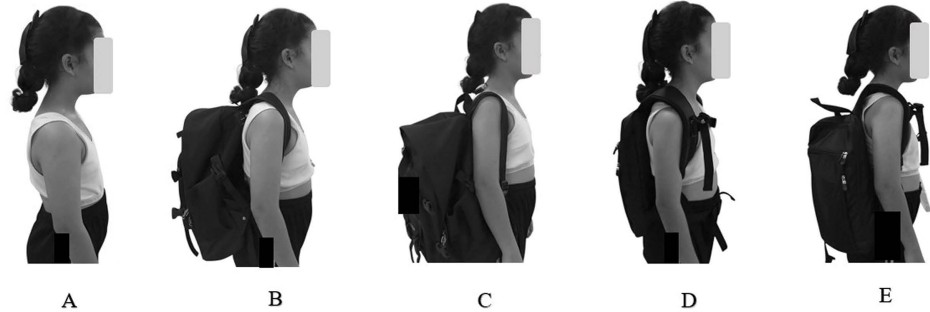

**Fig 3. Type of carrying bag. (A)** No bag. **(B)** Carrying a bilateral traditional bag. **(C)** Carrying a unilateral traditional bag. **(D)** = Carrying a bilateral modified bag. **(E)** Carrying a unilateral modified bag.

## Result

### Anthropometric data

Thirty-two healthy primary students (17 males and 15 females) participated in the study. The average age was $9.00 \pm 1.66$ years, and BMI was $18.12 \pm 3.48$ kg/m$^2$. Anthropometric data were measured for nine dimensions. The average shoulder-elbow height, sitting shoulder height, and sitting elbow height were $25.97 \pm 4.11$, $36.98 \pm 10.44$, and $26.35 \pm 3.50$ cm,

respectively. The average hip breadth and shoulder breadth were 26.13±11.73 and 31.12±4.05 cm. Moreover, chest and waist circumference were 55.08±18.46 and 54.65±19.10 cm. There were no differences between the boys and girls in anthropometric data, Table 1.

## Physiological responses

A two-way repeated measures ANOVA was conducted to examine the effects of bag type and carrying method, as well as their interaction, on the outcome variables. The $\%SpO_2$, heart rate, respiratory rate, systolic blood pressure, and diastolic blood pressure were assessed immediately pre- and post-walking under each condition. A statistically significant main effect was seen between the traditional and modified bags for heart rate ($p=0.001$, $\eta p^2=0.316$), respiratory rate ($p=0.007$, $\eta p^2=0.214$), systolic blood pressure ($p<0.001$, $\eta p^2=0.387$), and diastolic blood pressure ($p=0.002$, $\eta p^2=0.279$). In addition, a statistically significant main effect was seen between the one-strap and two-strap for heart rate ($p=0.014$, $\eta p^2=0.178$), Table 2.

There were no statistically significant interaction effects between carrying methods (unilateral vs bilateral) and bag type (traditional vs modified) on any physiological parameters, including oxygen saturation ($p=0.356$, $\eta p^2=0.028$), heart rate ($p=0.069$, $\eta p^2=0.102$), respiratory rate ($p=0.850$, $\eta p^2=0.001$), systolic blood pressure ($p=0.721$, $\eta p^2=0.004$), and diastolic blood pressure ($p=0.401$, $\eta p^2=0.023$). Mean $SpO_2$ values remained consistently high across all conditions, ranging from 98.78±2.17% to 99.47±0.84%. Heart rate was lower in the modified bag conditions than in the traditional bag, particularly with bilateral application (87.81±12.27 vs 92.47±16.05 beats/min), although this difference did not reach statistical significance. Respiratory rate and both systolic and diastolic blood pressure showed only minor variations across conditions, as shown in Table 3.

## Forward head-neck posture

A statistically significant main effect difference was observed between the traditional and modified bag in neck tilt ($p<0.001$, $\eta p^2=0.350$), head tilt ($p=0.001$, $\eta p^2=0.324$), and forward head posture ($p<0.001$, $\eta p^2=0.335$), respectively. Furthermore, a statistically significant main effect difference in head tilt between the one-strap and two-strap configurations ($p=0.021$, $\eta p^2=0.159$), Table 4.

**Table 1. The participant's anthropometric data ($n=32$).**

| Parameter | Mean±SD | | | p-value |
| --- | --- | --- | --- | --- |
| | Male (n=17) | Female (n=15) | Total (n=32) | |
| Age (yrs) | 9.47±1.84 | 8.40±1.24 | 9.00±1.66 | 0.943 |
| Weight (kg) | 32.76±8.38 | 33.16±12.08 | 32.95±10.11 | 0.512 |
| Height (cm) | 133.99±15.38 | 132.72±14.02 | 133.39±14.53 | 0.231 |
| BMI (kg/m$^2$) | 18.13±3.35 | 18.11±3.73 | 18.12±3.48 | 0.552 |
| Shoulder-elbow height (cm) | 25.96±4.00 | 25.99±4.36 | 25.97±4.11 | 0.231 |
| Sitting shoulder height (cm) | 39.78±5.60 | 33.80±13.61 | 36.98±10.44 | 0.625 |
| Sitting elbow height (cm) | 26.21±3.61 | 26.50±3.50 | 26.35±3.50 | 0.169 |
| Hip breadth (cm) | 27.53±15.74 | 24.55±4.07 | 26.13±11.73 | 0.256 |
| Shoulder breadth (cm) | 31.72±4.08 | 30.43±4.05 | 31.12±4.05 | 0.438 |
| Chest circumference (cm) | 59.74±17.22 | 49.80±18.96 | 55.08±18.46 | 0.900 |
| Waist circumference (cm) | 58.39±17.04 | 50.40±20.96 | 54.65±19.10 | 0.280 |

Data are presented as mean and standard deviation. Statistical significance was determined using the t- test. Abbreviation: BMI, body mass index.

**Table 2. Main effects of bag type (traditional vs modified) and carry method (unilateral vs bilateral) on physiological parameters (n = 32).**

| Parameters | Bag type | | p-value | ηp² | Carrying method | | p-value | ηp² |
|---|---|---|---|---|---|---|---|---|
| | Traditional bags | Modified bag | | | One strap | Two straps | | |
| SpO₂ (%) | 99.25±0.12 | 98.80±0.21 | 0.078 | 0.097 | 99.13±0.22 | 98.92±0.14 | 0.435 | 0.020 |
| Heart rate (bpm) | 95.94±2.26 | 88.63±11.92 | 0.001* | 0.316 | 94.42±1.76 | 90.14±2.28 | 0.014* | 0.178 |
| Respiratory rate (bpm) | 31.59±0.86 | 26.88±1.90 | 0.007* | 0.214 | 30.61±1.66 | 27.86±1.18 | 0.076 | 0.098 |
| Systolic blood pressure (mmHg) | 108.33±1.30 | 103.17±1.49 | <0.001** | 0.387 | 106.45±1.09 | 105.05±1.73 | 0.376 | 0.025 |
| Diastolic blood pressure (mmHg) | 78.05±1.64 | 74.13±1.55 | 0.002* | 0.279 | 77.11±1.76 | 75.06±1.51 | 0.147 | 0.066 |

Data are presented as mean and standard deviation. Statistical significance was determined using the repeated measures ANOVA test. *p-value is significant at the 0.05 level, **p-value is significant at the 0.001 level. ηp² = partial eta-squared.

**Table 3. Physiological responses across combined backpack conditions and interaction effects between bag type and carrying method (n = 32).**

| Parameters | Unilateral traditional bag | Bilateral traditional bag | Unilateral modified bag | Bilateral modified bag | Interaction p-value | ηp² |
|---|---|---|---|---|---|---|
| SpO₂ (%) | 99.47±0.84 | 99.03±0.97 | 98.78±2.17 | 98.81±1.23 | 0.356 | 0.028 |
| Heart rate (bpm) | 99.41±14.14 | 92.47±16.05 | 89.44±10.39 | 87.81±12.27 | 0.069 | 0.102 |
| Respiratory rate (bpm) | 33.13±5.09 | 30.06±7.00 | 28.09±7.09 | 25.66±8.27 | 0.850 | 0.001 |
| Systolic blood pressure (mmHg) | 109.22±7.92 | 107.44±11.78 | 103.69±7.42 | 102.66±9.37 | 0.721 | 0.004 |
| Diastolic blood pressure (mmHg) | 79.69±12.00 | 76.41±10.56 | 74.53±10.46 | 73.72±9.40 | 0.401 | 0.023 |

Data are presented as mean and standard deviation. Statistical significance was determined using the repeated measures ANOVA test. ηp² = partial eta-squared.

**Table 4. Main effects of bag type (traditional vs modified) and carrying methods (unilateral vs bilateral) on forward head-neck posture (n = 32).**

| Parameters | Bag type | | p-value | ηp² | Carrying methods | | p-value | ηp² |
|---|---|---|---|---|---|---|---|---|
| | Traditional bags | Modified bag | | | One strap | Two straps | | |
| Neck tilt (°) | 49.75±1.05 | 54.55±1.33 | <0.001* | 0.350 | 51.85±1.15 | 52.45±1.14 | 0.522 | 0.013 |
| Head tilt (°) | 78.60±1.56 | 73.14±1.02 | 0.001** | 0.324 | 77.83±1.56 | 73.91±1.14 | 0.021* | 0.159 |
| Forward head posture (cm) | 8.68±4.59 | 7.49±0.39 | <0.001** | 0.335 | 8.18±0.44 | 7.99±0.39 | 0.405 | 0.022 |

Data are presented as mean and standard deviation. Statistical significance was determined using the repeated measures ANOVA test. *p-value is significant at the 0.05 level, **p-value is significant at the 0.001 level. ηp² = partial eta-squared

A statistically significant interaction effect between carrying method (unilateral vs bilateral) and bag type (traditional vs modified) on neck tilt ($p < 0.001$, $\eta p^2 = 0.336$), head tilt ($p = 0.038$, $\eta p^2 = 0.132$), Table 5.

The pairwise comparisons showed that when carrying the traditional bag, a smaller neck tilt was seen compared to carrying the modified bag (49.75°±1.05°, $p < 0.001$). For head tilt and forward head posture, carrying the traditional bag had a greater head tilt (78.60°±1.56°, $p = 0.001$) and forward head posture (8.679±4.59 cm, $p < 0.001$) than carrying the modified bag. Moreover, carrying with one strap had a greater head tilt than when carrying with two straps (77.83°±1.56°, $p = 0.021$), Table 5.

Post hoc pairwise comparisons showed significant differences between several conditions (Table 6). During neck tilt, the modified bag showed greater degrees than the traditional bag under bilateral (mean difference = −2.831, $p = 0.015$, 95% CI: −5.379 to −0.283) and unilateral conditions (mean difference = −6.756, $p < 0.001$, 95% CI: −9.944 to −3.567).

**Table 5. Forward head–neck posture across combined backpack conditions and interaction effects between bag type and carrying method (n=32).**

| Parameters | Unilateral traditional bag | Bilateral traditional bag | Unilateral modified bag | Bilateral modified bag | Interaction p-value | ηp² |
|---|---|---|---|---|---|---|
| Neck tilt (°) | 48.47±1.34 | 51.03±1.16 | 58.14±1.65 | 53.86±1.43 | <0.001** | 0.336 |
| Head tilt (°) | 81.97±12.78 | 75.23±9.47 | 73.69±7.17 | 72.59±7.44 | 0.038* | 0.132 |
| Forward head posture (cm) | 8.94±3.10 | 8.42±2.34 | 7.42±2.48 | 7.56±2.25 | 0.146 | 0.067 |

Data are presented as mean and standard deviation. Statistical significance was determined using the repeated measures ANOVA test. **p-value is significant at the 0.001 level. ηp² = partial eta-squared.

Neck tilt was lower in the unilateral traditional bag condition compared with the bilateral traditional bag condition (mean difference = 2.559, $p = 0.034$), while modified conditions did not differ ($p = 0.104$).

Head tilt was greater in the unilateral traditional bag condition compared with the unilateral modified bag condition (mean difference = 8.281, $p < 0.001$, 95% CI: 4.438 to 12.125). Under the same traditional bag condition, head tilt was greater with unilateral carrying compared with bilateral carrying (mean difference = −6.741, $p < 0.010$, 95% CI: −11.763 to −1.718), Table 6.

## Discussion

This study investigated the physiological responses and head–neck posture associated with different backpack designs and carrying methods in primary school students. The results demonstrated that the modified bag design produced significantly lower heart rate, blood pressure, and respiratory rate compared with the traditional bag. Additionally, improvements in head and neck posture were observed when the modified design was used. These findings suggest that backpack design characteristics and load-distribution strategies play an important role in reducing physiological strain and postural deviations in schoolchildren. This was consistent with the results of previous research demonstrating that heavy backpacks can elevate heart rate and systolic blood pressure due to increased physical demand and altered posture [38,39]. The elevated respiratory rate observed in the current study when carrying traditional bags may be attributed to postural compensation and reduced lung capacity, which aligns with previous findings [40]. Although the load in the current study was standardized at 10% of body weight for both backpack conditions, the modified backpack was designed with reduced thickness, allowing the load to be positioned closer to the trunk. This likely reduced the load's moment arm relative to the spinal axis, thereby decreasing the mechanical demand on the cervical and thoracic regions. Previous biomechanical studies have demonstrated that load position, rather than load magnitude alone, significantly influences postural adaptation and physiological responses during load carriage. When the backpack is positioned farther from the body, the external flexion moment increases, requiring greater muscular activation to maintain upright posture. In contrast, reducing the load-to-body distance minimizes torque and may explain the lower physiological responses observed in the modified backpack condition [6,38].

The results of this study indicated a decrease in neck tilt and an increase in head tilt when participants carried the traditional bag. The neck tilt decreased by approximately 5° when participants carried the traditional bag (49.75° ± 1.05°) compared to the modified bag (54.55° ± 1.33°). This finding aligns with that of Ellapen et al., who reported a mean craniovertebral angle of 53.9° ± 14.6° when standing without a backpack, decreasing to 50.4° ± 16.4° when carrying a loaded backpack (P < 0.05) [9]. Prolonged neck flexion is a risk factor for neck pain. In neutral posture, a consistent anterior shear force of approximately 18 N is observed at the cervical joint level. During flexion, compressive forces throughout the cervical spine approximately double, whereas anterior shear forces increase fourfold in the upper cervical segments and decrease markedly in the lower (caudal) regions. These substantial alterations in joint kinetics provide mechanical

**Table 6. The pairwise comparisons in the forward head-neck posture of students.**

| Parameters | Parameters | Mean difference | p-value | 95%CI | |
|---|---|---|---|---|---|
| | | Degrees (°) | | Lower bound | Upper bound |
| **Neck tilt** | | | | | |
| Bilateral traditional bag | Bilateral modified bag | −2.831 | 0.015* | −5.379 | −0.283 |
| Unilateral traditional bag | Unilateral modified bag | −6.756 | <0.001** | −9.944 | −3.567 |
| Bilateral traditional bag | Unilateral traditional bag | 2.559 | 0.034* | −0.198 | 5.317 |
| Bilateral modified bag | Unilateral modified bag | −1.366 | 0.104 | −3.531 | 0.799 |
| **Head tilt** | | | | | |
| Bilateral traditional bag | Bilateral modified bag | 2.641 | 0.188 | −1.361 | 6.643 |
| Unilateral traditional bag | Unilateral modified bag | 8.281 | <0.001** | 4.438 | 12.125 |
| Bilateral traditional bag | Unilateral traditional bag | −6.741 | <0.010* | −11.763 | −1.718 |
| Bilateral modified bag | Unilateral modified bag | −1.100 | 0.496 | −4.354 | 2.154 |

Statistical significance was determined using the paired t- test. The mean difference is significant at the 0.05 level. **$p$-value = 0.001.

evidence supporting the influence of posture and its interaction with muscular activity in the pathogenesis of chronic neck pain [41]. Carrying a traditional bag promotes increased neck flexion, which, when sustained over time, may elevate the risk of developing neck pain compared to carrying a modified bag. Moreover, the results indicated that head tilt measured 78.60 ± 1.56° when carrying a traditional bag, reflecting an increase of approximately 5° compared to carrying a modified bag. Additionally, forward head posture increased approximately 1 cm when using a traditional bag. This finding aligns with those of Ramadan and Al-Tayyar, who reported that forward head posture without a bag was 3.1 ± 0.62 cm, while with a bag it was 5.6 ± 1.17 cm, showing a significant difference between forward head posture with and without a backpack ($p < 0.05$) [42]. As forward head posture increases, the C7 marker moves anteriorly. The closer the points at the shoulder and C7, the larger the sagittal shoulder angle [43]. Therefore, the more anterior head position observed in most participants in this study while carrying a backpack may have contributed to an increase in the sagittal shoulder angle [43]. These alterations in neck alignment can lead to strain on the cervical joints and soft tissues, as well as muscle function imbalance. This may result in pain in the cervical, upper thoracic, and shoulder areas [44].

Importantly, a significant interaction between bag type and carrying method was found for neck and head posture. This indicates that the effect of the modified bag is not independent of how the load is carried. Post hoc results showed that the modified bag reduced neck and head tilt more clearly under unilateral carrying, whereas these differences were less evident under bilateral carrying. In contrast, within the traditional bag condition, unilateral carrying resulted in greater head tilt than bilateral carrying. These findings suggest that the ergonomic advantage of the modified bag is maximized under asymmetric load conditions, while bilateral carrying may reduce postural differences between designs by improving load symmetry and postural stability. In contrast, no interaction effects were observed for physiological outcomes, despite significant main effects of bag type. This suggests that physiological responses are mainly driven by overall load characteristics rather than the combination of design and carrying method.

The design features of the modified backpack likely contributed to the observed benefits. First, the bag thickness was reduced compared to the traditional design, resulting in a lower capacity for storing textbooks and likely minimized the load moment arm. Consequently, the bag exerted less pressure on the back and lower portions of the body, even when carrying the same weight as a traditional bag. Oxygen consumption, minute ventilation, heart rate, discomfort scale, and rating of perceived exertion are notably reduced when the bag is positioned higher (around T7) compared to when it is placed lower (around T12 or L3) [45,46]. Higher backpack positioning results in improved outcomes, including a smaller change in craniovertebral angle and reduced upper trapezius pain [47]. Second, the bag strap was designed to be wider.

Wider straps exert significantly lower pressure on both the shoulder and strap areas [18]. Third, the inclusion of chest and waist straps likely improved load stabilization by limiting excessive movement of the backpack relative to the trunk. When both straps were fastened, the bag remained closer to the body, preventing excessive forward or backward leaning. whereas straps worn around the waist for abdominal support can notably reduce forward leaning [17]. Fourth, the modified bag features a rubber foam design integrated into the shoulder straps and areas of the bag that come into contact with the student's back. Double-layered straps, consisting of a reinforced internal support layer and a soft outer layer, reduce subclavian artery strain by 40% and skin stress by 50% compared with standard softer straps [48].

The findings of the current study also demonstrated that carrying with one strap resulted in greater heart rate and head tilt compared to carrying with two straps. This is supported by the findings of Daniel et al., who suggested that asymmetrical loading, such as using one strap, places uneven stress on the musculoskeletal system and requires greater muscular effort to maintain balance and posture [49]. Carrying a load on one side increases trunk and spinal muscle activity to counterbalance the asymmetry, resulting in higher energy expenditure and elevated heart rate. This reflects the greater physical effort required to maintain stability and gait symmetry under conditions of asymmetrical loading [50]. This study also highlighted that the one-strap carriage caused noticeable changes in head and neck tilt, allowing for compensation of the shifted center of gravity. While this helps maintain visual and balance alignment, it can lead to cervical strain over time, increasing the risk of neck discomfort and long-term spinal issues [51]. In contrast, using two straps allows for symmetrical weight distribution across the shoulders and back, reducing muscular strain and maintaining proper alignment of the head and spine. This method is generally recommended by health professionals for schoolchildren and adults who frequently carry backpacks [52]. Overall, the findings of this study suggest that backpack design features—rather than a specific product alone—play a critical role in influencing both physiological responses and head–neck posture during load carriage. The observed main effects indicate that improved design can reduce physiological stress, while the significant interaction effects highlight that postural outcomes are also dependent on how the load is carried. In particular, design characteristics such as reduced load-to-body distance, wider shoulder straps, stabilization straps, and bilateral load carriage may enhance postural alignment and contribute to safer backpack use among primary school students.

## Conclusion

This study examined the physiological responses and head–neck posture associated with different backpack designs and carrying methods in primary school students. Although the load was standardized at 10% of body weight, significant differences were observed between backpack conditions. The modified design resulted in lower heart rate, blood pressure, respiratory rate, and body discomfort, as well as reduced deviations in head and neck posture compared with the traditional backpack. These differences are likely attributable to design-related factors affecting load placement and stabilization rather than load magnitude alone. Specifically, positioning the load closer to the trunk and improving load distribution may reduce mechanical demand and compensatory postural adjustments. Additionally, carrying the backpack using two shoulder straps resulted in more favorable physiological and postural outcomes compared with unilateral carrying, highlighting the importance of symmetrical load distribution. Overall, the findings suggest that backpack design characteristics and carrying strategies play a critical role in minimizing physiological strain and postural deviation in schoolchildren.

## Acknowledgments

The authors express their gratitude to all the participants involved in the study. They also extend their appreciation to the directors and teachers of the participating primary schools for their invaluable support. We also extend our sincere gratitude to Prof. Dr. James Richards, School of Health, Social Work, and Sport, University of Central Lancashire, United Kingdom, for his valuable advice and consultation throughout this research.

## Author contributions

**Conceptualization:** Jittaporn Mongkonkansai, Udomsak Narkkul, Chadapa Rungruangbaiyok, Chuchard Punsawad.

**Data curation:** Jittaporn Mongkonkansai, Udomsak Narkkul, Chadapa Rungruangbaiyok, Chuchard Punsawad.

**Formal analysis:** Jittaporn Mongkonkansai, Udomsak Narkkul, Chadapa Rungruangbaiyok, Chuchard Punsawad.

**Funding acquisition:** Jittaporn Mongkonkansai, Chuchard Punsawad.

**Investigation:** Jittaporn Mongkonkansai, Udomsak Narkkul, Chadapa Rungruangbaiyok, Chuchard Punsawad.

**Methodology:** Jittaporn Mongkonkansai, Udomsak Narkkul, Chadapa Rungruangbaiyok, Chuchard Punsawad.

**Project administration:** Jittaporn Mongkonkansai, Udomsak Narkkul, Chadapa Rungruangbaiyok, Chuchard Punsawad.

**Validation:** Jittaporn Mongkonkansai, Udomsak Narkkul, Chadapa Rungruangbaiyok, Chuchard Punsawad.

**Writing – original draft:** Jittaporn Mongkonkansai, Udomsak Narkkul, Chadapa Rungruangbaiyok, Chuchard Punsawad.

**Writing – review & editing:** Jittaporn Mongkonkansai, Udomsak Narkkul, Chadapa Rungruangbaiyok, Chuchard Punsawad.

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
