## [Decision Letter · Decision Letter 0]

16 Feb 2026

PONE-D-25-54379Evaluating physiological responses and forward head-neck posture in primary students in Thailand when carrying nano and traditional backpacksPLOS One

Dear Dr. Punsawad,

Thank you for submitting your manuscript to PLOS ONE. After careful consideration, we feel that it has merit but does not fully meet PLOS ONE’s publication criteria as it currently stands. Therefore, we invite you to submit a revised version of the manuscript that addresses the points raised during the review process.

We look forward to receiving your revised manuscript.

Kind regards,

Tomoyoshi Komiyama, Ph.D

Academic Editor

PLOS One

Journal Requirements:

“This project was funded by the National Science and Technology Development Agency (NSTDA) of Thailand and the Walailak University Graduate Research Fund (CGS-RF-2023/04).”

4. We notice that your supplementary figures are uploaded with the file type 'Figure'. Please amend the file type to 'Supporting Information'. Please ensure that each Supporting Information file has a legend listed in the manuscript after the references list.

Additional Editor Comments:

Dear authors,

Your study investigated the effects of traditional versus nano backpacks on physiological responses, head–neck posture, and discomfort in primary school children. Overall, nano bags were associated with better posture, lower physiological stress, and less discomfort, suggesting a safer alternative to reduce long-term musculoskeletal risk in children.

However, I think that it is necessary to strengthen the reliability of these results by adding as much information as possible.

We thus have some questions and suggestions for the manuscript that you might consider. I believe these comments will be helpful in the revision of your study.

Tomoyoshi Komiyama

Reviewer's Responses to Questions

**Comments to the Author**

1. Is the manuscript technically sound, and do the data support the conclusions?

Reviewer #1: Yes

Reviewer #2: Yes

Reviewer #3: Yes

2. Has the statistical analysis been performed appropriately and rigorously? 

Reviewer #1: Yes

Reviewer #2: No

Reviewer #3: Yes

3. Have the authors made all data underlying the findings in their manuscript fully available?

Reviewer #1: Yes

Reviewer #2: Yes

Reviewer #3: Yes

4. Is the manuscript presented in an intelligible fashion and written in standard English?

Reviewer #1: Yes

Reviewer #2: Yes

Reviewer #3: Yes

5. Review Comments to the Author

Reviewer #1: The paper does make a significance contribution on the field. Well done to researchers. These findings will assist scientists to develop strategies to combat the ill effects of backpack loading among children.

The paper is well written. The good flow regarding what the investigation aimed to review and the discussion of the findings.

I recommend paper for publication.

Reviewer #2: Overall, this manuscript represents a technically sound evaluation of the effects of a common backpack design in comparison to an intentionally-modified alternative, with unilateral vs bilateral carriage comparisons. To some degree, this nested design is redundant, given the plethora of previous literature indicating the effects of unilateral backpack carriage, but the results do not detract from the larger narrative and provide some validating overlap with previous publications.

I have two primary concerns with the manuscript in its current state:

(1) The use of the term "nano" bag only suggests a difference in size. Readers may therefore misunderstand the nuanced differences upon finding this manuscript in databases. I would recommend abandoning the "nano" moniker in the title for something more informative (or at least less singular in implication).

(2) The statistical methodology is somewhat unusual, due to the immediate assumption of a lack of normal distribution, and only pairwise analysis despite nested conditions (traditional vs nano and bilateral vs unilateral). The Wilcoxon test is conservative, undermining effective power due to the non-normal assumption, which may be unnecessary. I would recommend testing for normality, and then seriously considering a repeated measures ANOVA in order to determine the effects of each condition more precisely while controlling for multiple tests.

Tertiary comments:

- I appreciate the comprehensive, yet brief review in the introduction section.

- On line 165, you describe recording blood pressure measurements at 5-minute intervals, but describe 1-minute intervals on line 167.

- In the first paragraph of the discussion section, you describe potential effects associated with load carriage. This is somewhat confusing, as it implies a difference in load between the two backpacks, yet your methodology describes that these were both fixed at 10% body weight. Is it perhaps more valuable to explore the potential effects of the decreased moment arm of the load in the nano backpack due to the decreased thickness?

- The conclusion ultimately recommends the nano backpack design; I agree that results suggest this conclusion, but I recommend that the discussion section is used to better explore the design considerations that likely led to these positive outcomes, and then for the conclusion to recommend specific bag features instead of a product. I would also recommend including recommendations related to unilateral vs bilateral load carriage, given that it was a component of the study design and analysis.

Reviewer #3: This study compared nano and traditional bags in children aged 7–12 years. Nano bags were associated with lower physiological stress and less head–neck posture deviation than traditional bags, regardless of how they were worn. These results suggest that nano bags may reduce musculoskeletal risk during prolonged use. However, I have a few minor comments, which are outlined below.

There are slight discrepancies in the numerical values reported between the Results and the Discussion sections.

For example, regarding head tilt: in the Results (Lines 318–319 and Table 4), the values are 77.50° for the traditional bag and 72.85° for the nano bag (a difference of 4.65°), whereas in the Discussion (Lines 368–369), the head tilt with the traditional bag is reported as 78.60 ± 1.56° and described as an increase of approximately 5° compared with the nano bag.

This discrepancy may be due to the use of parametric statistics (e.g., mean ± standard deviation) in the Discussion. However, since the Methods section states that medians and interquartile ranges were used for analysis (Line 233), unifying the reported values to medians throughout the manuscript would help avoid reader confusion and improve statistical consistency.

In addition, in the Discussion of forward head posture, the manuscript states: “Additionally, forward head posture increased by 1 to 2 cm when using a traditional bag.” Given that the reported medians are 8.13 cm and 7.14 cm (a difference of 0.99 cm), it may be more consistent with the description of head tilt to express this as “approximately 1 cm.”

6. PLOS authors have the option to publish the peer review history of their article (what does this mean?). If published, this will include your full peer review and any attached files.

Reviewer #1: No

Reviewer #2: No

Reviewer #3: No

---

## [Author Response · Author response to Decision Letter 1]

10 Mar 2026

Point-by-point responses to Reviewers' comments

Reviewer #1: The paper does make a significance contribution on the field. Well done to researchers. These findings will assist scientists to develop strategies to combat the ill effects of backpack loading among children.

The paper is well written. The good flow regarding what the investigation aimed to review and the discussion of the findings.

I recommend paper for publication.

Response: Thank you.

Reviewer #2: Overall, this manuscript represents a technically sound evaluation of the effects of a common backpack design in comparison to an intentionally-modified alternative, with unilateral vs bilateral carriage comparisons. To some degree, this nested design is redundant, given the plethora of previous literature indicating the effects of unilateral backpack carriage, but the results do not detract from the larger narrative and provide some validating overlap with previous publications.

I have two primary concerns with the manuscript in its current state:

(1) The use of the term "nano" bag only suggests a difference in size. Readers may therefore misunderstand the nuanced differences upon finding this manuscript in databases. I would recommend abandoning the "nano" moniker in the title for something more informative (or at least less singular in implication).

Response: Thank you for the suggestion. We have replaced the term “nano bag” with “modified bag” throughout the manuscript and highlighted the differences between the bags to reduce any misunderstanding.

(2) The statistical methodology is somewhat unusual, due to the immediate assumption of a lack of normal distribution, and only pairwise analysis despite nested conditions (traditional vs nano and bilateral vs unilateral). The Wilcoxon test is conservative, undermining effective power due to the non-normal assumption, which may be unnecessary. I would recommend testing for normality, and then seriously considering a repeated measures ANOVA in order to determine the effects of each condition more precisely while controlling for multiple tests.

Response: We tested and found the data to be normally distributed. We have therefore rerun the analysis using a Repeated Measures ANOVA (Line 244-305).

Tertiary comments:

- I appreciate the comprehensive, yet brief review in the introduction section.

Response: Thank you.

- On line 165, you describe recording blood pressure measurements at 5-minute intervals, but describe 1-minute intervals on line 167.

Response: Apologies for this inconsistency. We have edited this and report 5 minute intervals on line 166.

- In the first paragraph of the discussion section, you describe potential effects associated with load carriage. This is somewhat confusing, as it implies a difference in load between the two backpacks, yet your methodology describes that these were both fixed at 10% body weight. Is it perhaps more valuable to explore the potential effects of the decreased moment arm of the load in the nano backpack due to the decreased thickness?

Response: Thank you for this observation. We have changed the first paragraph of the discussion section and have added information on the potential effects of the decreased moment arm of the load in the nano backpack due to the decreased thickness in lines 308-329 as you suggested.

- The conclusion ultimately recommends the nano backpack design; I agree that results suggest this conclusion, but I recommend that the discussion section is used to better explore the design considerations that likely led to these positive outcomes, and then for the conclusion to recommend specific bag features instead of a product. I would also recommend including recommendations related to unilateral vs bilateral load carriage, given that it was a component of the study design and analysis.

Response: We have adjusted the discussion and conclusion. This now recommends specific bag features, and also includes recommendations related to unilateral vs bilateral load carriage in the conclusion lines 396-410 as you suggested.

Reviewer #3: This study compared nano and traditional bags in children aged 7–12 years. Nano bags were associated with lower physiological stress and less head–neck posture deviation than traditional bags, regardless of how they were worn. These results suggest that nano bags may reduce musculoskeletal risk during prolonged use. However, I have a few minor comments, which are outlined below.

There are slight discrepancies in the numerical values reported between the Results and the Discussion sections.

For example, regarding head tilt: in the Results (Lines 318–319 and Table 4), the values are 77.50° for the traditional bag and 72.85° for the nano bag (a difference of 4.65°), whereas in the Discussion (Lines 368–369), the head tilt with the traditional bag is reported as 78.60 ± 1.56° and described as an increase of approximately 5° compared with the nano bag.

This discrepancy may be due to the use of parametric statistics (e.g., mean ± standard deviation) in the Discussion. However, since the Methods section states that medians and interquartile ranges were used for analysis (Line 233), unifying the reported values to medians throughout the manuscript would help avoid reader confusion and improve statistical consistency.

Response: We tested and found the data to be normally distributed. We have therefore rerun the analysis using a Repeated Measures ANOVA (Line 244-305). These values have been checked and should now be consistent throughout the manuscript.

In addition, in the Discussion of forward head posture, the manuscript states: “Additionally, forward head posture increased by 1 to 2 cm when using a traditional bag.” Given that the reported medians are 8.13 cm and 7.14 cm (a difference of 0.99 cm), it may be more consistent with the description of head tilt to express this as “approximately 1 cm.”

Response: Thank you for your observation. We have edited this sentence, “Additionally, forward head posture increased approximately 1 cm when using a traditional bag” in line 346.

---

## [Decision Letter · Decision Letter 1]

30 Mar 2026

PONE-D-25-54379R1Evaluating physiological responses and forward head-neck posture in primary students in Thailand when carrying modified and traditional backpacksPLOS One

Dear Dr. Punsawad,

Thank you for submitting your manuscript to PLOS ONE. After careful consideration, we feel that it has merit but does not fully meet PLOS ONE’s publication criteria as it currently stands. Therefore, we invite you to submit a revised version of the manuscript that addresses the points raised during the review process.

We look forward to receiving your revised manuscript.

Kind regards,

Tomoyoshi Komiyama, Ph.D

Academic Editor

PLOS One

Journal Requirements:

Additional Editor Comments:

Dear authors,

Thank you for submitting your revised manuscript.

I think it is an improvement from the previous version.

However, one reviewer had additional questions.

This reviewer noticed some important information was missing,

so their judgement is major rather than minor revision.

Please answer these questions as listed below.

Tomoyoshi Komiyama

Reviewers' comments:

Reviewer's Responses to Questions

**Comments to the Author**

1. If the authors have adequately addressed your comments raised in a previous round of review and you feel that this manuscript is now acceptable for publication, you may indicate that here to bypass the “Comments to the Author” section, enter your conflict of interest statement in the “Confidential to Editor” section, and submit your "Accept" recommendation.

Reviewer #2: All comments have been addressed

Reviewer #3: All comments have been addressed

2. Is the manuscript technically sound, and do the data support the conclusions?

Reviewer #2: Yes

Reviewer #3: Partly

3. Has the statistical analysis been performed appropriately and rigorously? 

Reviewer #2: Yes

Reviewer #3: No

4. Have the authors made all data underlying the findings in their manuscript fully available?

Reviewer #2: Yes

Reviewer #3: No

5. Is the manuscript presented in an intelligible fashion and written in standard English?

Reviewer #2: Yes

Reviewer #3: Yes

6. Review Comments to the Author

Reviewer #2: I am satisfied with the revisions to the manuscript and appreciate the authors' efforts in the revision process.

Reviewer #3: Dear Authors,

Thank you very much for submitting your revised manuscript. However, I would greatly appreciate it if you could kindly address the following additional questions.

While the use of analysis of variance (ANOVA) is appropriate assuming the normality of the data, there are several critical issues regarding the reporting of statistical results and the presentation of data.

1. Statistical Analysis

The study employs repeated measures ANOVA; however, only the main effects are reported,

and the interaction effect (Bag type × number of straps) is missing. The authors should include the interaction statistics in the results and discuss whether the modified bag’s benefits depend on the carrying method.

2. Table Column Configuration

The study compares four specific conditions derived from the combination of Bag type (Traditional vs. Modified [Nano]) and Number of straps (Unilateral vs. Bilateral). In the current version (Revision 1), Tables 2, 3, 5, and 6 present "Traditional" and "Modified" in Columns 1 and 2, and "One strap" and "Two straps" in Columns 3 and 4. This layout is confusing as it appears to separate the factors rather than showing the combined experimental conditions. Considering the experimental conditions, it would be better to present the Mean and SD using the column configuration from the original version.

3. Inconsistencies in Statistical Reporting (P-values)

There is a highly concerning inconsistency in the reporting of p-values. The p-values presented in Table 2 and Table 3 for physiological responses are identical. Similarly, the p-values in Table 5 and Table 6 for head-neck posture are also identical. Statistically, the p-value for a Main Effect in a repeated measures ANOVA and the p-value for a post-hoc pairwise comparison should not be identical. This suggests that the results from the ANOVA may have been simply copied into the post-hoc tables, or that the post-hoc analysis was not performed correctly. Please clarify this and provide the correct, adjusted p-values for all pairwise comparisons.

4. Missing Table 4

The manuscript currently skips from Table 3 to Table 5. I suspect this is simply a clerical error in numbering, as there is no mention of a Table 4 in the text. Please renumber the subsequent tables and update the citations in the text.

7. PLOS authors have the option to publish the peer review history of their article (what does this mean?). If published, this will include your full peer review and any attached files.

Reviewer #2: No

Reviewer #3: No

---

## [Author Response · Author response to Decision Letter 2]

20 Apr 2026

A point-by-point response to the reviewer’s comments

Review Comments to the Author

Reviewer #2: I am satisfied with the revisions to the manuscript and appreciate the authors' efforts in the revision process.

Response: Thank you.

Reviewer #3: Dear Authors,

Thank you very much for submitting your revised manuscript. However, I would greatly appreciate it if you could kindly address the following additional questions.

While the use of analysis of variance (ANOVA) is appropriate assuming the normality of the data, there are several critical issues regarding the reporting of statistical results and the presentation of data.

1. Statistical Analysis

The study employs repeated measures ANOVA; however, only the main effects are reported,

and the interaction effect (Bag type × number of straps) is missing. The authors should include the interaction statistics in the results and discuss whether the modified bag’s benefits depend on the carrying method.

Response: We thank the reviewer for this important suggestion. The Results section has been revised to include the interaction effects between bag type and carrying method (Bag type × number of straps) in Table 3 and 5. The findings now report significant interaction effects for head–neck posture variables, while no significant interactions were observed for physiological outcomes.

In addition, the Discussion section has been updated to address these findings. We clarify that the benefits of the modified backpack on postural outcomes depend on the carrying method, with more pronounced improvements observed under unilateral carriage, whereas differences between bag designs were reduced during bilateral carriage. This revision strengthens the interpretation of our findings and highlights the importance of considering both ergonomic design and carrying strategy (lines 262–332 and 387–398).

2. Table Column Configuration

The study compares four specific conditions derived from the combination of Bag type (Traditional vs. Modified [Nano]) and Number of straps (Unilateral vs. Bilateral). In the current version (Revision 1), Tables 2, 3, 5, and 6 present "Traditional" and "Modified" in Columns 1 and 2, and "One strap" and "Two straps" in Columns 3 and 4. This layout is confusing as it appears to separate the factors rather than showing the combined experimental conditions. Considering the experimental conditions, it would be better to present the Mean and SD using the column configuration from the original version.

Response: We thank the reviewer for this helpful suggestion. In the revised manuscript, we have improved the clarity of data presentation by reporting the four combined experimental conditions (bilateral traditional, unilateral traditional, bilateral modified, and unilateral modified) in the interaction tables (Tables 3, 5, and 6), which directly reflect the 2×2 factorial design. At the same time, we have retained Table 2 and 4 to present the main effects of bag type and carrying method derived from the two-way repeated measures ANOVA. This was done to facilitate interpretation of the independent contribution of each factor, which may not be immediately apparent from the condition-specific comparisons alone. Therefore, the current presentation includes both factor-level (main effects) and condition-level (combined conditions) results, providing a more comprehensive and interpretable view of the findings while remaining consistent with the study design in lines 262-332.

3. Inconsistencies in Statistical Reporting (P-values)

There is a highly concerning inconsistency in the reporting of p-values. The p-values presented in Table 2 and Table 3 for physiological responses are identical. Similarly, the p-values in Table 5 and Table 6 for head-neck posture are also identical. Statistically, the p-value for a Main Effect in a repeated measures ANOVA and the p-value for a post-hoc pairwise comparison should not be identical. This suggests that the results from the ANOVA may have been simply copied into the post-hoc tables, or that the post-hoc analysis was not performed correctly. Please clarify this and provide the correct, adjusted p-values for all pairwise comparisons.

Response: Thank you for carefully highlighting this issue. We agree that the previously reported p-values were inconsistent and may have caused confusion. Upon re-examination, all post-hoc analyses were re-conducted. The pairwise comparisons were performed using paired t-tests for predefined contrasts, and the p-values were recalculated accordingly.

In the present study, we performed post-hoc pairwise comparisons based on a priori hypotheses, rather than conducting all possible pairwise comparisons among the four conditions (which would result in six comparisons).

Specifically, we focused on four predefined comparisons that directly addressed the study objectives:

(1) comparisons between bag types within the same carrying method (bilateral traditional vs bilateral modified; unilateral traditional vs unilateral modified), and

(2) comparisons between carrying methods within the same bag type (bilateral vs unilateral for traditional and modified bags).

The revised tables (Tables 6) now present corrected mean differences, 95% confidence intervals, and adjusted p-values. Importantly, the post-hoc p-values are no longer identical to those from the repeated measures ANOVA and accurately reflect the statistical differences between specific conditions.

These corrections have been incorporated into the revised manuscript (lines 262-332).

4. Missing Table 4

The manuscript currently skips from Table 3 to Table 5. I suspect this is simply a clerical error in numbering, as there is no mention of a Table 4 in the text. Please renumber the subsequent tables and update the citations in the text.

Response: I have renumbered the subsequent tables and updated the citations in the text in line 290-291.

---

## [Decision Letter · Decision Letter 2]

10 May 2026

Evaluating physiological responses and forward head-neck posture in primary students in Thailand when carrying modified and traditional backpacks

PONE-D-25-54379R2

Dear Dr. Punsawad,

We’re pleased to inform you that your manuscript has been judged scientifically suitable for publication and will be formally accepted for publication once it meets all outstanding technical requirements.

Kind regards,

Tomoyoshi Komiyama, Ph.D

Academic Editor

PLOS One

Additional Editor Comments (optional):

Dear Authors,

Thank you for submitting your revised manuscript.

I found the revised version much easier to understand than the original manuscript.

I am satisfied with both the authors’ responses and the revisions made to the manuscript, and I am happy to recommend its acceptance.

The authors have addressed the remaining comments from all three reviewers satisfactorily.

Therefore, I have no further comments, and all of my previous concerns have been adequately resolved.

I believe this manuscript will be of considerable interest to readers.

Tomoyoshi Komiyama

Reviewers' comments:

Reviewer's Responses to Questions

**Comments to the Author**

1. If the authors have adequately addressed your comments raised in a previous round of review and you feel that this manuscript is now acceptable for publication, you may indicate that here to bypass the “Comments to the Author” section, enter your conflict of interest statement in the “Confidential to Editor” section, and submit your "Accept" recommendation.

Reviewer #3: All comments have been addressed

2. Is the manuscript technically sound, and do the data support the conclusions?

Reviewer #3: Yes

3. Has the statistical analysis been performed appropriately and rigorously? 

Reviewer #3: Yes

4. Have the authors made all data underlying the findings in their manuscript fully available?

Reviewer #3: Yes

5. Is the manuscript presented in an intelligible fashion and written in standard English?

Reviewer #3: Yes

6. Review Comments to the Author

Reviewer #3: Dear Authors,

I would like to commend the authors for their thorough revisions. The manuscript has been significantly improved through the inclusion of interaction effects and the reorganization of the tables, which have greatly enhanced the clarity and rigor of the analysis. The authors have fully addressed my concerns, and I believe the paper is now ready for publication.

7. PLOS authors have the option to publish the peer review history of their article (what does this mean?). If published, this will include your full peer review and any attached files.

Reviewer #3: No

---

## [Editor Report · Acceptance letter]

PONE-D-25-54379R2

PLOS One

Dear Dr. Punsawad,

I'm pleased to inform you that your manuscript has been deemed suitable for publication in PLOS One. Congratulations! Your manuscript is now being handed over to our production team.

Kind regards,

on behalf of

Dr. Tomoyoshi Komiyama

Academic Editor

PLOS One